# Low Temperature Synthesis of High-Density Carbon Nanotubes on Insulating Substrate

**DOI:** 10.3390/nano9030473

**Published:** 2019-03-21

**Authors:** Ying Xiao, Zubair Ahmed, Zichao Ma, Changjian Zhou, Lining Zhang, Mansun Chan

**Affiliations:** 1Department of Electronic and Computer Engineering, Hong Kong University of Science and Technology, Kowloon, Hong Kong, China; zahmed@connect.ust.hk (Z.A.); zmaaa@connect.ust.hk (Z.M.); mchan@ust.hk (M.C.); 2School of Electronic and Information Engineering, South China University of Technology, Guangzhou 510640, China; zhoucj@scut.edu.cn; 3College of Electronic Science and Technology, Shenzhen University, Shenzhen 518061, China; eelnzhang@szu.edu.cn

**Keywords:** CNT growth, low temperature, insulating substrate, CMOS compatible

## Abstract

A method to synthesize high-density, vertically-aligned, multi-wall carbon nanotubes (MWCNTs) on an insulating substrate at low temperature using a complementary metal–oxide–semiconductor (CMOS) compatible process is presented. Two factors are identified to be important in the carbon nanotube (CNT) growth, which are the catalyst design and the substrate material. By using a Ni–Al–Ni multilayer catalyst film and a ZrO_2_ substrate, vertically-aligned CNTs can be synthesized at 340 °C using plasma-enhanced chemical vapor deposition (PECVD). Both the quality and density of the CNTs can be enhanced by increasing the synthesis temperature. The function of the aluminum interlayer in reducing the activation energy of the CNT formation is studied. The nanoparticle sintering and quick accumulation of amorphous carbon covering the catalyst can prematurely stop CNT synthesis. Both effects can be suppressed by using a substrate with a high surface energy such as ZrO_2_.

## 1. Introduction

The unique properties of carbon nanotubes (CNTs) have enabled many applications in modern electronic technology, such as contact electrodes and active film for sensors, memory, transistors and membranes [1,2,3,4]. Although CNT synthesis is relatively mature, new constraints are imposed when they are integrated into existing applications. In particular, one promising application for CNTs is to enhance the performance of complementary metal–oxide–semiconductor (CMOS) interconnect technology [5,6,7,8]. To be compatible with the CMOS back-end process, the maximum processing temperature must be lower than 450 °C [5] and the catalyst used cannot cause contamination to the underlying devices [6]. To provide maximum flexibility for integrating CNTs in different regions of the interconnect layers, methods to synthesize CNTs on both the interconnect metal and the interlayer dielectric are needed. A CMOS compatible process to synthesize high-density CNTs on an insulating substrate can further enable CNTs to be used in sensor, memory and transistor applications [9,10,11,12].

Chemical vapor deposition (CVD) has been widely used for large-scale synthesis of high quality CNTs due to its controllability and low process temperature [13,14]. Synthesis of high-density vertical CNTs on conductive and/or insulating substrates with Fe, Ni or Co catalysts has already been achieved at temperatures ranging from 550 °C to 950 °C [15,16,17]. By using plasma-enhanced chemical vapor deposition (PECVD), researchers have successfully produced high-density, vertically-aligned CNTs on conductive substrates with Ni or Co catalysts at around 450 °C, as the energetic plasma enables feed gas dissociation at reduced temperatures [18,19,20]. For insulating substrates, an Fe catalyst is often used to synthesize CNTs at low temperatures [21,22,23]. However, Fe is not CMOS compatible due to contamination issues [5,8]. A new catalyst has to be designed for such a purpose.

With the given background, a study is conducted to demonstrate a CMOS compatible process in synthesizing high-density CNTs on an insulating substrate at a temperature below 450 °C. Based on this process, we further investigate the following: (1) the tradeoff between CNT properties when growing CNTs at a low temperature on an insulating substrate; (2) the effect of a catalyst structure using CMOS-compatible material; and (3) the impact of the substrate material on the CNT synthesis process. The results of this study are reported in the subsequent sections of this paper.

## 2. Experiment and Results 

To synthesize CNTs with CMOS-compatible techniques, a 10 nm ZrO_2_ layer was first deposited on N-type (100) silicon using atomic layer deposition (ALD) to form the insulating substrate. A specially-designed Ni–Al–Ni (1 nm–0.5 nm–1 nm) multilayer catalyst was deposited by electron-beam (E-beam) evaporation with a deposition rate of 0.5 Å/s, under a chamber pressure maintained at 1.5 × 10^−6^ Torr. Three wafers were then transferred to a PECVD chamber (Seki Technotron Corp.AX5200M). To enhance the density of CNTs, these wafers were annealed at 450 °C, 400 °C and 340 °C, respectively, with 3:1 composition of H_2_ and N_2_ at 2.8 Torr to form uniform nanoparticles. To start the CNT synthesis process in the PECVD chamber, 30 sccm methane (CH_4_) was introduced for 3, 4 and 6 minutes under 200 W plasma power for CNT growth at 450 °C, 400 °C and 340 °C, respectively. The wafers were then cooled down to room temperature in H_2_ atmosphere.

Figure 1a–c shows the scanning electron microscope (SEM, JEOL-6700) images of the CNT forest grown in the three different temperatures of 450 °C, 400 °C and 340 °C. It is observed that vertically-aligned CNTs were successfully synthesized at all three temperatures. While the synthesis was successful at 340 °C, the grown CNTs had the shortest length and the lowest density of around 1 × 10^10^ tubes/cm^2^. The CNT density was obtained by counting the CNTs over a given area from an SEM image and then dividing the number of the corresponding area. To reduce random error due to process variations, 10 samples were averaged for each data point. The density of the CNTs grown at 400 °C and 450 °C were around 1.21 × 10^11^ tubes/cm^2^ and 5 × 10^11^ tubes/cm^2^, respectively. 

As it was reported that the properties of the synthesized CNTs are highly correlated with the catalyst structure [20], the resulting physical geometries of the catalyst annealed at different temperatures are shown in Figure 1d–f. It was found that higher annealing temperatures produce more isolated nanoparticles that prevented sintering during the CNT synthesis process. As a result, higher density and better aligned CNTs were achieved at higher temperatures. More effects of the catalyst structure on the properties of synthesized CNTs will be discussed in the next section. Besides the catalyst, the temperature also affects the CNT growth rate. The CNT growth rates at 340 °C, 400 °C and 450 °C were 300 nm/min, 500 nm/min and 1.2 µm/min, respectively. A higher growth rate usually associates with higher quality, higher density and better vertical alignment of the synthesized CNTs [24].

The transmission electron microscope (TEM) images of CNTs synthesized at 340 °C and 450 °C obtained by the JEOL 2010 system are shown in Figure 2. The images reveal that most of the catalysts are located at the top of the CNTs. This implies that the adhesion between the catalyst and the substrate is weak, resulting in CNT synthesis in tip-growth mode. The TEM images also reveal the layered structure, indicating that the CNTs are multi-wall carbon nanotubes (MWCNTs). From the samples grown at 450 °C, the diameter of the CNTs ranges from 8 nm to 58 nm, as shown in Figure 3, with an average diameter of 26 nm. 

To characterize the CNT quality, Raman spectroscopy was used with a 514 nm laser excitation, and the results are shown in Figure 4. The Raman spectrum shows that the G-peak (due to bond stretching between pairs of sp^2^ carbon atoms) and the D-peak (due to breathing modes of sp^2^ carbon atoms in rings) are located at around 1604 cm^−1^ and 1340 cm^−1^ respectively. Since the D-band in sp^2^ carbon is usually observed when the crystal symmetry is broken by defects (usually vacancy or line dislocation), it is used to characterize the degree of disorder in carbon nanotubes. The quality of the CNTs is characterized by the ratio of the intensity of the D-peak to G-peak (I_D_/I_G_). As shown in Figure 4, the measured I_D_/I_G_ of the synthesized CNTs at 340 °C, 400 °C and 450 °C are 0.85, 0.82 and 0.81, respectively. The I_D_/I_G_ obtained are all smaller than those reported by a similar previous study performed at a low temperature [24]. It suggests that the defects of the synthesized CNTs using the method described can achieve fewer defects when compared with some existing studies [24]. It is also discovered that higher growth temperatures can reduce defects and improve the quality of the synthesized CNTs.

## 3. Discussion and Analysis

The Ni–Al–Ni multilayer catalyst is a key factor that facilitates CNT synthesis at a low temperature on an insulating substrate, by reducing the activation energy of the reaction. Figure 5 shows the growth rate of CNTs at different temperatures, which follows the form of the Arrhenius equation. The activation energy for CNT growth, based on the Arrhenius equation, can be obtained from the slope of Figure 5 and is estimated to be 0.35 eV. Compared to the activation energy for CNT growth using the Fe catalyst (0.56 eV) and Co–Al catalyst (0.40 eV) [24], this value is about 0.21 eV and 0.05 eV less, respectively. This is the reason why the Ni–Al–Ni multilayer catalyst allows high-density CNT synthesis at a temperature as low as 340 °C. The low activation energy of the Ni–Al–Ni multilayer catalyst makes it easier to disassociate the carbon–hydrogen source and supply sufficient carbon atoms at low temperatures. 

In addition to lowering the activation energy, the aluminum in the Ni–Al–Ni multilayer catalyst can also prevent the catalyst from sintering on the substrates, making the spread of the catalyst nanoparticles more uniform in the CNT synthesis process [20,24]. When pure Ni catalyst is used, the strong atomic attraction will cause the nanoparticles to join together, leading to non-uniform distribution of the nanoparticles. With the Al layer, the Ni–Al compound nanoparticle has a lower mobility which immobilizes the nanoparticle on the ZrO_2_ surface and thus suppresses sintering of the catalyst on the substrate. Hence, it is more difficult for the nanoparticles to merge together and form a large catalyst cluster, thereby maintaining the uniformity of the nanoparticles. However, when the Al layer in the multilayer catalyst is too thick, the Al atoms can sufficiently cover the Ni atoms blocking the catalyst action of Ni as well as causing sintering due to the attraction of the Al atoms. Figure 6a–c shows the atomic force microscopy (AFM) image of the nanoparticles formed using different catalyst film composition. The sample with Ni–Al–Ni catalyst of 0.5 nm-thick aluminum shows the highest catalyst nanoparticle density when compared with the nanoparticles formed using pure Ni or Ni–Al–Ni multiple layers with 1 nm aluminum. Figure 6d–f show the SEM images of the CNTs grown at 450 °C using Ni–Al–Ni catalysts with different Al thicknesses. It is observed that the longest and densest vertically-aligned CNTs are synthesized from the Ni–Al–Ni catalyst with 0.5 nm thick Al. The growth rate and density of synthesized CNTs using different catalysts are summarized in Table 1. When the Al is too thick, it actually degrades the growth rate, the quality, and the density of the synthesized CNTs compared to the case of pure Ni catalyst. Therefore, the thickness of the Al layer is very critical when synthesizing high quality and high-density CNTs.

Compared to growing CNTs on a metallic substrate, growing CNTs on an insulator does not have the problem of the catalyst diffusing into the metal, as the catalyst usually has weak interactions with the substrate [24,25]. However, the weak adhesion between the catalyst and the substrate may result in the migration of the nanoparticle catalyst on the surface, causing catalyst sintering during the synthesis process. This is particularly true for insulating substrates with low surface energy, like SiO_2_ (0.29 J/m^2^) and Al_2_O_3_ (0.64 J/m^2^). To prevent catalyst sintering, a substrate with a high surface energy should be used for CNT synthesis. Therefore, ZrO_2_ with a surface energy of 1.08 J/m^2^ is selected in this study. 

In addition to preventing nanoparticle catalyst sintering, the strong interaction between the ZrO_2_ substrate and the catalyst reduces the activeness of the catalyst and slows down the formation of amorphous carbon in the beginning of the CNT synthesis [26,27]. This prevents the surface of the catalyst from being quickly covered by the nucleated amorphous carbon. The extended exposure of the catalyst nanoparticles allow them to be transformed into an elongated shape when the catalyst atoms are pushed up by the newly formed CNT underneath. This facilitates the alignment of the graphene layers into a cylindrical structure [27].

Figure 7 shows the results of CNTs grown on 20 nm SiO_2_, and 20 nm Al_2_O_3_ with the same Ni–Al–Ni (1 nm–0.5 nm–1 nm) multilayer catalyst under the same synthesis conditions described earlier. There are no CNTs grown on the SiO_2_, as the catalyst was covered by the amorphous carbon before any CNTs could be synthesized. On the Al_2_O_3_ substrate, a small number of short CNTs can be observed, but the synthesis stopped quickly in the process due to the formation of amorphous carbon. Therefore, the choice of ZrO_2_ as a substrate is an important factor to enhance the CNT synthesis process on insulating substrates. 

## 4. Conclusions

We have demonstrated the synthesis of CNTs on an insulating substrate using a CMOS- compatible catalyst at low temperature. The use of a Ni–Al–Ni multilayer catalyst and ZrO_2_ substrate are two key factors to achieve high-density, vertically-aligned MWCNTs at a low temperature. The effects of the composition of the catalyst and the choice of substrate material have been studied based on the process developed. Furthermore, the mechanisms leading to the high-density synthesis at low temperature have been proposed. From the experimental results, a catalyst that provides a large reduction of the activation energy in the CNT growth process and a substrate material with a high surface energy are necessary for the success of the CNT synthesis. At a low temperature of 340 °C, we have successfully grown vertically-aligned CNTs with a density of 1 × 10^10^ tubes/cm^2^. Moreover, the quality and density can be enhanced at higher synthesis temperatures. Our results open new possibilities for CNTs, as a material of choice for next-generation CMOS-compatible applications.

## Figures and Tables

**Figure 1 nanomaterials-09-00473-f001:**
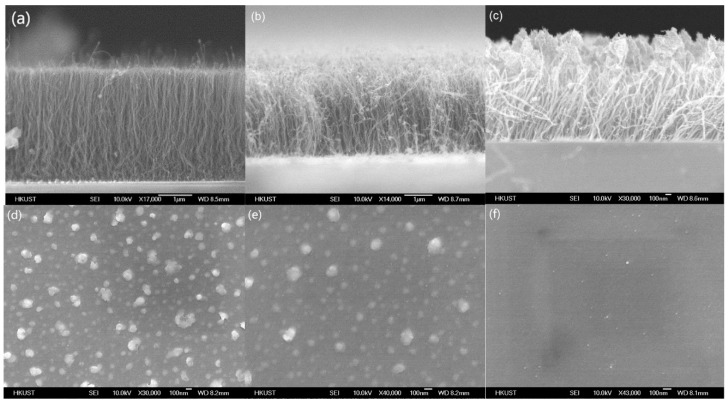
SEM images of vertically-aligned carbon nanotubes (CNTs) grown at (**a**) 450 °C, (**b**) 400 °C, and (**c**) 340 °C with a Ni–Al–Ni (1 nm–0.5 nm–1 nm) multilayer catalyst; and of nanoparticles annealed at (**d**) 450 °C, (**e**) 400 °C, and (**f**) 340 °C with the Ni–Al–Ni (1 nm–0.5 nm–1 nm) multilayer catalyst.

**Figure 2 nanomaterials-09-00473-f002:**
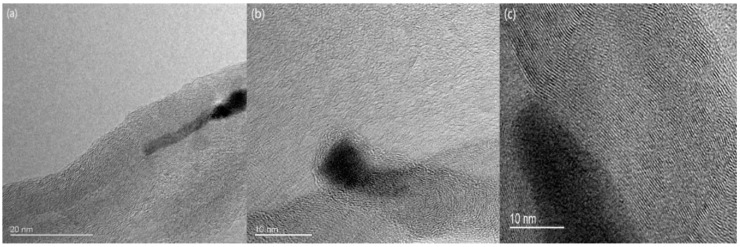
TEM images of the vertically-aligned CNTs, synthesized by Ni–Al–Ni (1 nm–0.5 nm–1 nm) catalyst. (**a**) Structure of the multi-wall carbon nanotubes (MWCNTs) grown at 340 °C, (**b**) the tip growth of the MWCNTs grown at 340 °C, and (**c**) structure of the MWCNTs grown at 450 °C.

**Figure 3 nanomaterials-09-00473-f003:**
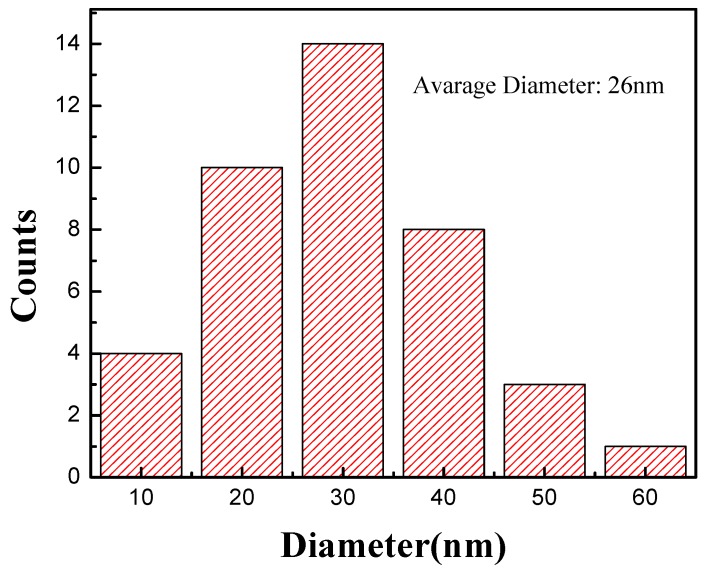
The diameter distribution of the vertically-aligned CNTs grown at 450 °C.

**Figure 4 nanomaterials-09-00473-f004:**
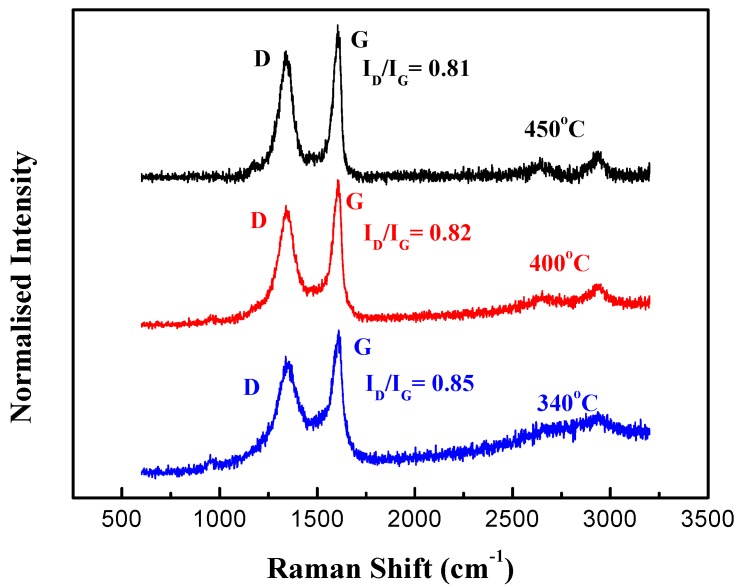
Raman spectrum of the vertically-aligned CNTs grown at 450 °C, 400 °C and 340 °C.

**Figure 5 nanomaterials-09-00473-f005:**
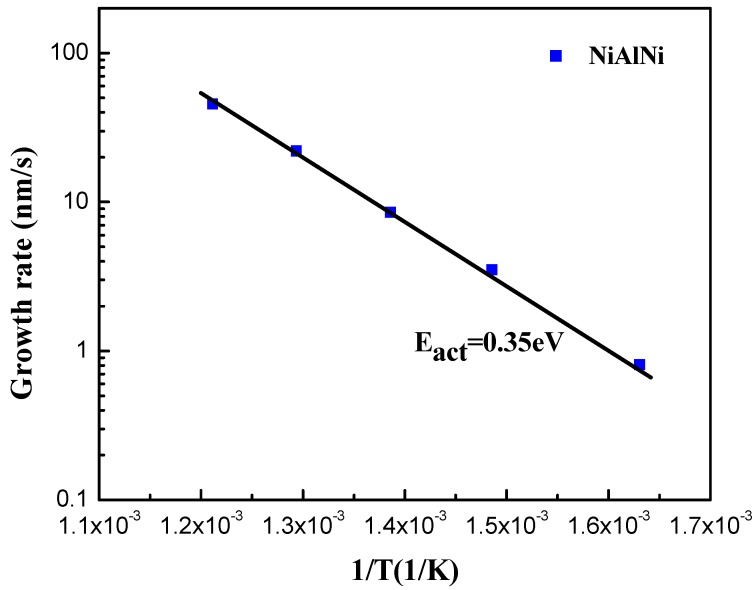
Arrhenius plot of growth rate versus inverse temperature of Ni–Al–Ni (1 nm–0.5 nm–1 nm) catalyst.

**Figure 6 nanomaterials-09-00473-f006:**
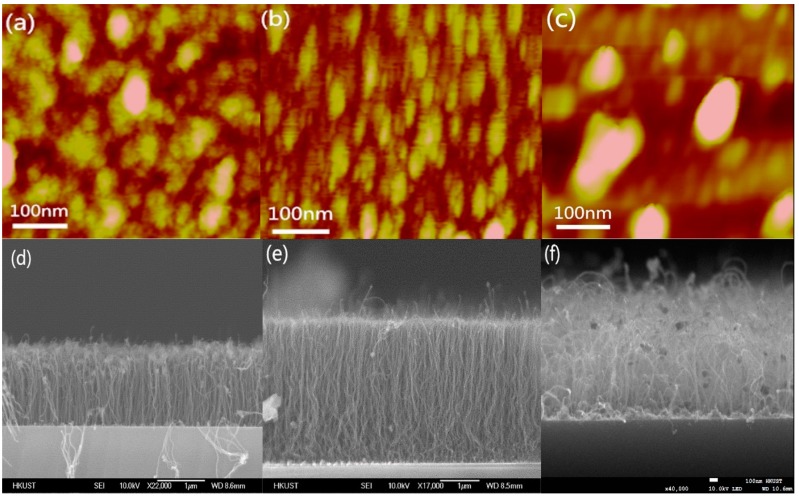
Atomic force microscopy (AFM) images (500 nm × 500 nm area) of the samples’ surface after annealing at 450 °C in N_2_ and H_2_ with (**a**) Ni (2 nm) catalyst, (**b**) Ni–Al–Ni (1 nm–0.5 nm–1 nm) multilayer catalyst, and (**c**) Ni–Al–Ni (1 nm-1 nm-1 nm) multilayer catalyst; and SEM images of vertically-aligned CNTs grown at 450 °C with (**d**) Ni (2 nm) catalyst, (**e**) Ni–Al–Ni (1 nm–0.5 nm–1 nm) catalyst, and (**f**) Ni–Al–Ni (1 nm–1 nm–1 nm) catalyst.

**Figure 7 nanomaterials-09-00473-f007:**
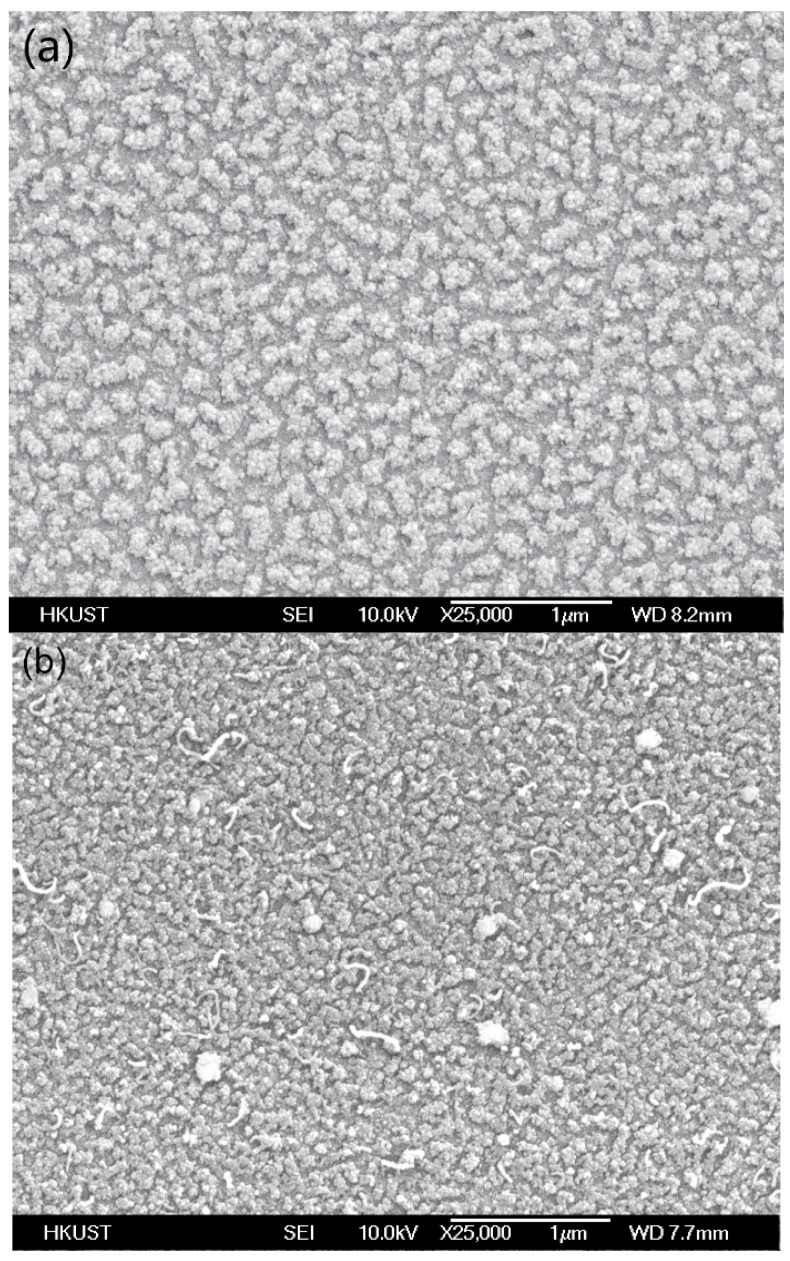
CNT synthesis results with (**a**) SiO_2_ and (**b**) Al_2_O_3_ with Ni–Al–Ni (1 nm–0.5 nm–1 nm) multilayer catalyst at 450 °C.

**Table 1 nanomaterials-09-00473-t001:** The growth rate and density of CNTs grown at 450 °C with different catalysts.

Catalysts	Growth Rate (nm/min)	Density (tubes/cm^2^)
Ni (2 nm)	800	1 × 10^11^
Ni–Al–Ni (1 nm–0.5 nm–1 nm)	1200	5 × 10^11^
Ni–Al–Ni (1 nm–1 nm–1 nm)	200	1 × 10^9^

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
