# Peer review of "Low Temperature Synthesis of High-Density Carbon Nanotubes on Insulating Substrate"

_nanomaterials, 2019, doi:10.3390/nano9030473_

Reviewer 1 Report

Very nice work!

Author Response

Thanks.

Reviewer 2 Report

1. There are many odd formulations which make the reading of the article very difficult.

 For example:

- Abstract, line 15: ”Two important factors are identified to be important …”. Recommend reformulation.  (“Two important factors are identified to be decisive …”, “Two factors are identified to be of tremendous importance …”).

- line 15: ”… which are …” (represented by …).

- line 38: “… people successfully produced …” (researchers, investigators).

- line 60: “…Ni-Al-Ni multilayer catalyst were designed …”(“was”, “has been designed for …” or another formulation: “Ni-Al-Ni multilayer catalysts were designed …”.

- line 60: “The multilayer catalyst were deposited …” (was).

- line 111: “…CNTs versus the growth temperature”. (…versus temperature growth)

2. The entire text must be revised from grammatical point of view and should be carefully checked. For example:

- line 140: “designing optimal thickness of aluminum is crucial to result in facilitates the synthesis of …” (Proposed reformulation: “designing optimal thickness of resulting aluminum is crucial and facilitates the synthesis of …”).

- line 136: “…The density of the MWCNTs with Ni-Al-Ni is catalyst around …” (Proposed reformulation: “The density of the MWCNTs with Ni-Al-Ni catalyst is around …”).

- line 135: “… which corresponding to the AFM results”.

3. line 74: “The density of the vertically aligned MWCNTs were estimated to be around …”. What was the procedure used for this estimation?

4. line 122: “…the addition of aluminum layer could prevent catalyst sintering to form high-density nanoparticles of catalyst”. This is a strong conclusion so the introduction of another reference will add value. 

5. line 180: “… carbon nanotube synthesis at ultralow temperature”. This expression: ”ultralow temperature” is usually associated with cryogenics, and it is not the case here, under the circumstances of this experiment.

Author Response

We would like to thank the reviewers for their constructive comments. We deeply appreciate the time that they have taken to review our work. These comments helped us to improve the quality of our manuscript. In addition, we would like to thank the reviewers and editors for the exceptional timeliness of their work.

Point-by-point responses to the reviewer’s comments are stated below.

1.  There are many odd formulations which make the reading of the article very difficult.

For example:

-  Abstract, line 15: ”Two important factors are identified to be important …”. Recommend reformulation.  (“Two important factors are identified to be decisive …”, “Two factors are identified to be of tremendous importance …”).

-  line 15: ”… which are …” (represented by …).

-    line 38: “… people successfully produced …” (researchers, investigators).

-    line 60: “…Ni-Al-Ni multilayer catalyst were designed …”(“was”,  “has been designed for …” or another formulation: “Ni-Al-Ni multilayer catalysts were designed …”.

-    line 60: “The multilayer catalyst were deposited …” (was).

-    line 111: “…CNTs versus the growth temperature”. (…versus temperature growth)

Response: The paper has been extensively revised to increase the readability. All the recommendations of the reviewers have been incorporated in the revision.

2.  The entire text must be revised from grammatical point of view and should be carefully checked. For example:

-    line 140: “designing optimal thickness of aluminum is crucial to result in facilitates the synthesis of …” (Proposed reformulation: “designing optimal thickness of resulting aluminum is crucial and facilitates the synthesis of …”).

-    line 136: “…The density of the MWCNTs with Ni-Al-Ni is catalyst around …” (Proposed reformulation: “The density of the MWCNTs with Ni-Al-Ni catalyst is around …”).

-    line 135: “… which corresponding to the AFM results”.

Response:  We have asked a native English speaker to check the paper and correct the mechanics of the English. The grammatical errors pointed out by the reviewer and many others have been corrected in the revised manuscript.

3.  line 74: “The density of the vertically aligned MWCNTs were estimated to be around …”. What was the procedure used for this estimation?

Response:  The density of MWCNT is commonly measured by weight-gain method or counting on SEM images. The weight-gain method is accurate when the CNTs are long and have very few defects. Counting the CNTs from an SEM image is a simpler method which is often used when the CNTs are short and have defects. In our work, the CNTs were grown at low temperature. They are relatively short and have defects. Therefore, we used an SEM image to count the number of CNTs to obtain its density. To reduce the effect from process variation, 10 samples were used to obtain an average value for each data point. The method to obtain the CNT density has been added [Line 72-74] in the revised manuscript.

4.  line 122: “…the addition of aluminum layer could prevent catalyst sintering to form high-density nanoparticles of catalyst”. This is a strong conclusion so the introduction of another reference will add value.

Response:  The statement is actually not a conclusion, but an explanation for the catalyst design based on existing knowledge. A reference [24] has been added to the revised manuscript to provide some background about the choice of catalyst material.

5.  line 180: “… carbon nanotube synthesis at ultralow temperature”. This expression: ”ultralow temperature” is usually associated with cryogenics, and it is not the case here, under the circumstances of this experiment.

Response:  We agree that the use of the term “ultra-low” is misleading.  The actually temperature (340℃) is used in the revised manuscript instead [line 192].

Reviewer 3 Report

Authors describe CNT(carbon fiberous structures) at 340degreeC.

research of this field is necessary.

authors claims 340 degree C growth of "CNTs".

but Figure 1 shows CNT like structures which holds a lot of amorphous carbon deposited on the top half area in SEM. is it CNTs? it look like carbon-carbon fiber composites.

authors claims 340 degreeC growth of CNTs and described in abstract, too.

but most data are of 450 degreeC, like Fig.6. it leads readers misunderstand.

L30 "However, ..."

Refer FUJITSU CNT works for LSI.

https://www.researchgate.net/publication/239663345_Electrical_Properties_of_Carbon_Nanotube_Bundles_for_Future_Via_Interconnects

L34-35, "As we know, ... CVD....CNTs due to good ..."

reference required.

L42

requires CMOS compatible Ni and Co catalyst growth of CNTs as reference.

L47, "but most reported works"

most means most.

refer the paper which describe cnt growth on insulator.

L67,

Growth time or feed duration of gas is required as CVD conditions.

L81 "structure under ..."

of which temperature?

L84

Comparison is necessary. It is known that CNTs are grown in tip growth mode by PE-CVD even on conductive substrates.

L88-L89

show data.

Fig2 requires growth temperature.

L102-L103

results on 400 degree C is required.

L 104 "fewer defects"

what is the definition of defects?

Figure 3 should have 400 degree C spectra.

Figure 4 needs unit in y axis.

L122-L124 needs data.

Figure 6 should be replaced with 340degree C SEM.

authors claims 340 degreeC growth of CNTs and described in abstract, too.

but most data are of 450 degreeC, like Fig.6. it leads readers misunderstand.

Figure 7 should be shown in same scale.

L180

what is definition of "ultralow"?

L166-170

thickness of SiO2, Al2O3 is required in experimental condition

Author Response

We would like to thank the reviewers for their constructive comments. We deeply appreciate the time that they have taken to review our work. These comments helped us to improve the quality of our manuscript. In addition, we would like to thank the reviewers and editors for the exceptional timeliness of their work.

Point-by-point responses to the reviewers’ comments are stated below.

1.  Authors describe CNT (carbon fiberous structures) at 340degreeC. Research of this field is necessary. authors claims 340 degree C growth of "CNTs". but Figure 1 shows CNT like structures which holds a lot of amorphous carbon deposited on the top half area in SEM. is it CNTs? it look like carbon-carbon fiber composites.

Response:  The CNTs grown at 340℃ has been confirmed to be MWCNT by TEM images and some of them are included in Figure 2.  We have clarified it in the revised manuscript [line 93-95]. The material deposited on the top was due to the continuous supply of feeding gas after PECVD turned off.    

2.  Authors claims 340 degreeC growth of CNTs and described in abstract, too. but most data are of 450 degreeC, like Fig.6. it leads readers misunderstand.

Response:  We have included data for CNT synthesized at 340oC, 400oC and 450oC in Fig. 1 and Fig. 3 when CNT synthesis is concerned. 340oC we mentioned in the abstract is just the lowest possible temperature we have successfully grow the CNT.  We have clarified it in the abstract. The purpose of Fig. 6 is to study the effect of catalyst structure rather than the temperature of CNT grow.  So, any temperature will do and we select 450oC because it create the best contrast among the different substrates.  The corresponding statement is rephrased in the revised manuscript. [line 144-150]. 

L30 "However, ..." Refer FUJITSU CNT works for LSI.

https://www.researchgate.net/publication/239663345_Electrical_Properties_of_Carbon_Nanotube_Bundles_for_Future_Via_Interconnects

Response: We have re-written the introduction and included a reference for the first demonstration of CNTs via [Ref. 2].  The work from LSI, FUJITSU CNT is also added in the reference list [Ref. 6] as an additional example of how CNT can be used in LSI applications.

L34-35, "As we know, ... CVD....CNTs due to good ..." reference required.

Response: Some references are added to describe the advantages of CVD method for CNT synthesis

in the revised manuscript. [Ref. 13, 14]

L42 requires CMOS compatible Ni and Co catalyst growth of CNTs as reference.

Response: References 18 and 20 are provided to explain the CMOS compatibility requirement.  

L47, "but most reported works" most means most. refer the paper which describe cnt growth on insulator.

Response: We have rephrased the sentence to remove the ambiguity. 

L67 Growth time or feed duration of gas is required as CVD conditions.

Response: Additional information including the growth time and temperature are included in the revised manuscript. [Line 65-66]

L81 "structure under ..."of which temperature?

Response: The temperature information is provided in the revised manuscript. [Line 94-95]

L84 Comparison is necessary. It is known that CNTs are grown in tip growth mode by PE-CVD even on conductive substrates.

Response: The sentence is revised to avoid any confusion [Line 95-97]. We have also provided further explanation about the resulting growth mechanism together with Figure 2.

L88-L89 show data.

Response: A new Figure 3 that shows the distribution of CNT diameter is added in the revised manuscript.

Fig2 requires growth temperature.

Response: The growth temperatures of CNTs are added to Fig. 2 in the revised manuscript.

L102-L103 results on 400 degree C is required.

Response: The results coming from 400℃ processing temperature are added in the revised manuscript accordingly. [Line 108-109]

L 104 "fewer defects" what is the definition of defects?

Response: The defects refer the line dislocation or vacancy.  The explanation is added in the revised manuscript. [Line 105-106].

Figure 3 should have 400 degree C spectra.

Response: The Raman spectra at 400℃ is included in figure 4 in the revised manuscript.

Figure 4 needs unit in y axis.

Response: The unit of the y-axis in figure 4 (Figure 5 in the revised manuscript) is added.

L122-L124 needs data.

Response: The conclusion is revised to clarify the confusion.  The low activation energy (0.35 eV) of the Ni-Al-Ni multilayer is stated in the revised manuscript. [Line 122-123]

Figure 6 should be replaced with 340degree C SEM. authors claims 340 degreeC growth of CNTs and described in abstract, too. but most data are of 450 degree C, like Fig.6. it leads readers misunderstand.

Response:   As explained in the response in the previous, we do not claim this work is for CNT growth at 340oC.  We just mention this number because it is the lowest temperature we are able to synthesize the CNT.  Higher temperature is needed for the quality requirement which has been clarified in the revised manuscript.  Figure 6 is to show the effect of catalyst engineering.  Due to the poor uniformity of the CNT synthesize at 340oC the effect of the catalyst engineering is not obvious.  And that is the reason we use CNT synthesize at 450oC to do the demonstration.  The corresponding statement is rephrased in the revised manuscript. [line 144-150]. 

Figure 7 should be shown in same scale.

ResponseFigure 7 is modified with SEM images of same scale in the revised manuscript.

L180 what is definition of "ultralow"?

Response: We have removed the description “low temperature” in the revision and exact value of is stated. [Line 194].

L166-170 thickness of SiO2, Al2O3 is required in experimental condition

Response: The exact thickness of SiO2 (20 nm) and Al2O3 (20 nm) are added in the revised manuscript. [Line 173]

Reviewer 4 Report

In this paper, the authors use PECVD to grow vertically aligned MWCNT on N-type silicon substrate. Ni-Al-Ni is used as catalyst and a layer of ZrO2 is deposited on the silicon substrate to increase adhesion. Although the paper is very well written, but it lacks any significant addition to the saturated field of the fabrication of CNTs using CVD methods. Based on that, I don’t recommend the publication of the paper in nanomterials

Author Response

We would like to thank the reviewers for their constructive comments. We deeply appreciate the time that they have taken to review our work. These comments helped us to improve the quality of our manuscript. In addition, we would like to thank the reviewers and editor for the exceptional timeliness of their work.

Point-by-point responses to the reviewers’ comments are stated below.

In this paper, the authors use PECVD to grow vertically aligned MWCNT on N-type silicon substrate. Ni-Al-Ni is used as catalyst and a layer of ZrO2 is deposited on the silicon substrate to increase adhesion. Although the paper is very well written, but it lacks any significant addition to the saturated field of the fabrication of CNTs using CVD methods. Based on that, I don’t recommend the publication of the paper in nanomaterials.

Response: Although there are lots of work about synthesis of CNT using CVD method such as reference 12-17. There are few works focusing providing CMOS compatible synthesis method to realize low temperature high density CNTs on insulators, which is necessary to achieve wide application of CNTs in transistor and sensor, capacitance and energy harvest devices. However, there are still some challenges in achieving low temperature, high quality growth of CNT on insulating substrates: (1) Low growth temperature slows the metal catalyst activation reaction, hindering dissociation of carbon feed gas and carbon nucleation, thus reducing CNT densities; (2) for CMOS technology requirement, Fe catalyst is contagious, which cannot be used for the CMOS technology. (3) weak adhesion strength between the catalyst and the substrate would cause nanoparticle sintering, and increase the possibility of catalyst poisoning by amorphous carbon [20-25].

To address these problem, in our work, a viable approach to grow vertically aligned MWCNT on insulating substrate at low temperature is presented, which opens new possibilities for CNTs, as material of choice in next-generation CMOS compatible sensor and device integrations. 

Reviewer 5 Report

General comments:

The authors describe a catalyst/substrate combination that enables MWCNT growth at very low temperatures, albeit at loss of crystallinity. The paper is well presented, a few comments to address are put below. It’s a neat short story and I recommend publication.

Language:

·       The authors should revisit the paper to insert the article ’the’ at multiple points in their text. I outline only very few in my corrections, but this is by far not meant as an all-including proof read.

·       Between values and units, it is custom to put a space.

·       Please review where to use CNTs or CNT. E.g. it should be CNT growth rather than CNTs growth.

Individual comments on phrasing / grammar / language:

15: important factors are identified to be important: repetition

20: ‘elongation’ is the process of lengthening something regarding a physical length. I assume the authors meant ‘increasing the lifetime of the catalyst’

42: there still remain_ challenges

122: What’s more, the addition of an aluminium layer could prevents catalyst sintering to form high-density catalyst nanoparticles, leading to high-density CNT_ growth, as we are showing below. The pure Ni catalyst would easily sinters on the substrate. With an aluminium layer, the multilayer catalyst prevents Ni from sintering on the insulating substrate.

131: Figure 6 shows the SEM images of CNTs grown at 450 C from Ni-Al-Ni catalysts with three different aluminium thicknesses.

135: which corresponds to the AFM results.

138: For _ CNTs grown from Ni-Al-Ni catalysts with a relatively thick aluminium layer…

140: Therefore, determining an optimal thickness…

158: ZrO2 layer with its higher surface energy (…) causes

160: ‘enhance the elongation’: the strong adhesion between substrate and catalyst helps to extend the lifetime of the catalyst and increases the tube formation time during synthesis.

With regards to content:

24: ‘in the last decade’: CNTs are famous at least since 1991 (Iijima) which is almost 30 years ago now, i.e. more than one decade. The authors should correct the time frame or specify which part of CNT science they refer to

27: ‘relatively mature in enabling monolithic 3D integration of memory and devices’: References needed to at least the most famous papers in this ‘mature field’

48: How does the electrical conductivity of the substrate influence CNT growth? To the CVD knowledgeable reader, the interaction with the catalyst material is obvious, as is the adhesion between substrate and catalyst which determines the agglomeration of catalyst droplets. However, the influence of substrate conductivity needs to be specified and supported with references

64: The authors should include an SEM of the catalyst nanoparticles formed at 340, 400 and 450 C. They are presumably primary to determine the type and quality of CNTs grown and their density. Together with figure 1, the authors should include a) an analysis of average size of catalyst as function of annealing temperature and b) average of walls as function of catalyst size or growth temperature

74: How was the density of MWCNTs determined?

Figure1: There is a clear degradation of CNT quality with decreasing temperature. The authors should comment on that, including their Raman of all three (!) samples, as function of temperature and regarding their CNTs’ usefulness for CMOS technology depending on defectivity.

84: ‘tip growth’: One advantage of wafer grown CNT forests is that in most cases it is base-growth and the catalyst stays on the substrate when the forest is peeled off. Comments on the CMOS usefulness when the catalyst stays in the CNTs?

87: Is the average number of walls the same for all growth temperatures?

106: Include Raman spectra of all three growth temperatures

11 / Figure4: The figure is not well explained in the text. a) How exactly is the activation energy determined/estimated? b) The points in the graph refer to temperatures approx. 850 – 1100 C. How do these points relate to the three temperatures discussed?

113: This activation energy is 0.21 eV and 0.05 eV less than that for MWCNT growth from Fe and Ci-Al catalyst, respectively.

117: Low activation energy and higher catalytic energy: Catalytic energy is neither stated nor referenced.

Figure4: Change position of the legend or make clear that it is a legend. At the moment it looks like an outlier of the data

125: Why is the electrical conductivity / insulating property important for sintering?

130: highest density of smaller sized catalyst?

Figure 5/6: The reviewer suggests combining figures 5 and 6 so that the AFM images are directly correlated with the SEMs of the same catalyst and growth conditions

160: How the adhesion helps preventing poisoning is not clear

Figure 7: As in my earlier comments, the connection between conductivity and adhesion is not explained here, however, the SEMs do show a clear difference which is interesting – although that cannot be causally related to the substrates conductivity. Please comment on the underlying physics of your statement

182: material of choice

Author Response

We would like to thank the reviewers for their constructive comments. We deeply appreciate the time that they have taken to review our work. These comments helped us to improve the quality of our manuscript. In addition, we would like to thank the reviewers and editors for the exceptional timeliness of their work.

Point-by-point responses to the reviewer’s comments are stated below.

The authors describe a catalyst/substrate combination that enables MWCNT growth at very low temperatures, albeit at loss of crystallinity. The paper is well presented, a few comments to address are put below. It’s a neat short story and I recommend publication.

Language: The authors should revisit the paper to insert the article ’the’ at multiple points in their text. I outline only very few in my corrections, but this is by far not meant as an all-including proof read.

· Between values and units, it is custom to put a space.

· Please review where to use CNTs or CNT. E.g. it should be CNT growth rather than CNTs growth.

Response:   We have asked a naïve English-speaking person to go through the paper to reduce the problem of the language. We hope the revised version of the manuscript will become more readable.

Individual comments on phrasing / grammar / language:

15:   important factors are identified to be important: repetition

20:   ‘elongation’ is the process of lengthening something regarding a physical length. I assume the authors meant ‘increasing the lifetime of the catalyst’

42:   there still remain_ challenges

122: What’s more, the addition of an aluminium layer could prevents catalyst sintering to form high-density catalyst nanoparticles, leading to high-density CNT_ growth, as we are showing below. The pure Ni catalyst would easily sinters on the substrate. With an aluminium layer, the multilayer catalyst prevents Ni from sintering on the insulating substrate.

131: Figure 6 shows the SEM images of CNTs grown at 450 C from Ni-Al-Ni catalysts with three different aluminium thicknesses.

135: which corresponds to the AFM results.

138: For _ CNTs grown from Ni-Al-Ni catalysts with a relatively thick aluminium layer…

140: Therefore, determining an optimal thickness…

158: ZrO2 layer with its higher surface energy (…) causes…

160: ‘enhance the elongation’: the strong adhesion between substrate and catalyst helps to extend the lifetime of the catalyst and increases the tube formation time during synthesis.

Response:  We appreciate the reviewer’s effort in providing detail suggestions on the language problem. All the recommendations have been incorporated in the revised manuscript.

With regards to content:

24:   ‘in the last decade’: CNTs are famous at least since 1991 (Iijima) which is almost 30 years ago now, i.e. more than one decade. The authors should correct the time frame or specify which part of CNT science they refer to

Response:  The timeframe is actually not necessary in the context of this paper.  To avoid the confusion, we decide to remove the reference on the duration of its popularity.

27:   ‘relatively mature in enabling monolithic 3D integration of memory and devices’: References needed to at least the most famous papers in this ‘mature field’.

Response:  This expression has been modified with the recommendation of the person who proofread this manuscript.  References [5-8] are added in the revised manuscript to provide some background about the popularity of CNT technology.

48:   How does the electrical conductivity of the substrate influence CNT growth? To the CVD knowledgeable reader, the interaction with the catalyst material is obvious, as is the adhesion between substrate and catalyst which determines the agglomeration of catalyst droplets. However, the influence of substrate conductivity needs to be specified and supported with references.

Response: The conductivity of substrates does not affect the CNT growth, but the properties of the catalyst during the CNT synthesis process.  Reference 20 and 24 helps to describe thte interaction between the catalyst with metallic substrate.  And reference 26 and 27 are added to provide the theory of how the surface energy of the substrate affect the CNT synthesis. [Line 160-167]

64:   The authors should include an SEM of the catalyst nanoparticles formed at 340, 400 and 450 C. They are presumably primary to determine the type and quality of CNTs grown and their density. Together with figure 1, the authors should include a) an analysis of average size of catalyst as function of annealing temperature and b) average of walls as function of catalyst size or growth temperature.

Response: The SEM images of catalyst nanoparticle formed at 340 ℃, 400℃ and 450 ℃ are added to Figure 1 in the revised manuscript.  The size of nanoparticle annealed at 450℃ is the largest.  The size of the nanoparticle decreases with temperature.  The discussion has been added in the revised manuscript. [Line 78-82]

74: How was the density of MWCNTs determined?

Response: The density of MWCNT is commonly measured by weight-gain method or counting on SEM images. The weight-gain method is accurate when the CNTs are long and have very few defects.  Counting the CNTs from an SEM image is a simpler method which is often used when the CNTs are short and have defects.  In our work, the CNTs were grown at low temperature.  They are relatively short and have defects.  Therefore, we used an SEM image to count the number of CNTs to obtain its density.  To reduce the effect from process variation, 10 samples were used to obtain an average value for each data point.  The method to obtain the CNT density has been added in the revised manuscript. [Line 74-77]

Figure1:   There is a clear degradation of CNT quality with decreasing temperature. The authors should comment on that, including their Raman of all three (!) samples, as function of temperature and regarding their CNTs’ usefulness for CMOS technology depending on defectivity.

Response: It is correct that the quality of the CNT degrades with decreasing temperature. This has been stated clearly in the revised manuscript [Line 81-84].  We have included the Raman spectra at all three different temperature being discussed in the manuscript.

84:   ‘tip growth’: One advantage of wafer grown CNT forests is that in most cases it is base-growth and the catalyst stays on the substrate when the forest is peeled off. Comments on the CMOS usefulness when the catalyst stays in the CNTs?

Response: Whether the it is desire to peel off the CNT or not is application specific.  In the applications we were working on that lead to this investigate, having the CNTs stay on the substrate is more desirable.  We have highlighted the targeted applications based on our earlier work in the revised manuscript. [Ref. 8]  

87: Is the average number of walls the same for all growth temperatures?

Response: The density of MWCNT depends on the size of catalyst nanoparticle, which is determined by the temperature.  Some discussion about it is given in the revised manuscript. [Line 77-80]

106: Include Raman spectra of all three growth temperatures

Response:  We have included the Raman spectra of samples grown at all 3 temperatures in Figure 4 of the revised manuscript. 

11 / Figure4:    The figure is not well explained in the text. a) How exactly is the activation energy determined/estimated? b) The points in the graph refer to temperatures approx. 850 – 1100 C. How do these points relate to the three temperatures discussed?

Response: The activation energy is calculated by the Arrhenius equation as below.

where k is the growth rate. T is the absolute temperature (in Kelvins). A is the pre-exponential factor, a constant for each chemical reaction.  Ea is the activation energy for the reaction. kB is the Boltzmann constant.  We take the natural logarithm of Arrhenius equation and plot it against experimental data obtained at 550, 500℃, 450 ℃, 400 ℃ and 340 ℃ which is shown in Figure 5.  The activation energy can then be obtained by the slope of the figure.  The explanation has been added in the revised manuscript. [Line 121-122]

113: This activation energy is 0.21 eV and 0.05 eV less than that for MWCNT growth from Fe and Ci-Al catalyst, respectively.

Response: We have revised this sentence accordingly. [Line 123-124]

117:    Low activation energy and higher catalytic energy: Catalytic energy is neither stated nor referenced.

Response: The “catalytic energy” has been remove in the revised manuscript.

Figure4:   Change position of the legend or make clear that it is a legend. At the moment it looks like an outlier of the data

Response: Figure 5 has been revised accordingly.

125: Why is the electrical conductivity / insulating property important for sintering?

Response: We agreed that there is some confusion in the original manuscript.  What affect the sintering process is the interaction between the catalyst and the substrate.  The substrate has two effects: (1) metallic substrate tends to absorb the catalyst while insulating substrate is inert; (2) the surface energy of the insulating substrate determines the potential migration during the sintering process.  We have clarify these points in the revised manuscript. [Line 159-166]

130: highest density of smaller sized catalyst?

Response: For the same catalyst, the smallest catalyst will cause highest density nanoparticle and CNTs at the same process condition. But for different catalyst, the smaller size of catalyst does not lead higher density nanoparticle. For example, pure 2 nm Ni does not have higher density CNTs than that of Ni-Al-Ni (1 nm-0.5 nm- 1 nm) multilayer catalyst due to sintering. [Line 143-146]

Figure 5/6:    The reviewer suggests combining figures 5 and 6 so that the AFM images are directly correlated with the SEMs of the same catalyst and growth conditions.

Response:The original figure 5 and 6 are combined as a new figure 6 as suggested.

160: How the adhesion helps preventing poisoning is not clear

Response: As pointed out by Reference 27, the strong adhesion between the ZrO2 substrate and the catalyst reduce the activeness of the catalyst that increase the time for CNT formation during the CNT synthesis process. This prevents the surface of the catalyst from being quickly covered by the nucleated amorphous carbon due to carbon accumulation.  The increase in synthesis also provide time for the catalyst to reshape into an elongated from that assists the alignment of graphene layers into a tubular structure. These explanations are added in revised manuscript. [Line 168-174]

Figure 7: As in my earlier comments, the connection between conductivity and adhesion is not explained here, however, the SEMs do show a clear difference which is interesting – although that cannot be causally related to the substrates conductivity. Please comment on the underlying physics of your statement

Response: As described in our earlier response, there is no direct relationship between the conductivity of the substrate and the catalyst adhesion.  It is just a co-incident that the catalyst is a metal, which has better adhesion to another metal that are conductors. We also admitted that the writing in the previous submission is a bit confusing.  The paper has been rewritten and we hope that the explanations on the CNT growth mechanism on different substrate in the revised manuscript is clearer. [Line 160-165]

182: material of choice

Response: The statement has been rephrased [Line 196].

Round  2

Reviewer 2 Report

I think that the material is modified, new figures (such as Fig.3 and Fig.4) have been added or revised, that English has been clearly verified and the content is significantly improved.

Reviewer 3 Report

Reviewers request are fully fulfilled by authors.

good work it is. 

Reviewer 4 Report

The reviewer still believes that the paper lacks any significant addition to the saturated field of the fabrication of CNTs using CVD methods. Based on that, I don’t recommend the publication of the paper in nanomterials

Reviewer 5 Report

The reviewer considers the changes to be made sufficiant and recommends publikation without further changes